# Deep Learning-Based Synthetic CT for Personalized Treatment Modality Selection Between Proton and Photon Therapy in Thoracic Cancer

**DOI:** 10.3390/cancers17091553

**Published:** 2025-05-03

**Authors:** Libing Zhu, Nathan Y. Yu, Riley C. Tegtmeier, Jonathan B. Ashman, Aman Anand, Jingwei Duan, Quan Chen, Yi Rong

**Affiliations:** 1Department of Radiation Oncology, Mayo Clinic, Phoenix, AZ 85058, USA; zhu.libing@mayo.edu (L.Z.);; 2Department of Radiation Oncology, University of South Florida Morsani College of Medicine, Tampa, FL 33606, USA; 3Department of Radiation Oncology, The University of Alabama at Birmingham, Birmingham, AL 35233, USA

**Keywords:** deep learning network, informed physician–patient consultation, synthetic CT, normal tissue complication probability

## Abstract

Selecting the most suitable radiation treatment modality is challenging before dedicated treatment planning scans are available, which is an issue commonly faced by radiation oncologists during patient consultations. An expedited treatment modality selection was proposed using deep learning-generated synthetic CT from an earlier diagnostic CT. By comparing normal tissue complication probabilities for critical organs, this study shows that the advantageous treatment modality chosen using synthetic CT matches with that on planning CT. This represents the first experience of using synthetic CT for guiding treatment modality selection for patients with tumors in the upper lung.

## 1. Introduction

Photon and proton beams are commonly used for radiotherapy of non-superficial tumors. Photon therapy is mainstream due to its clinical efficacy and lower cost [1,2]. Proton therapy, while more expensive, offers the advantage of minimizing radiation to healthy tissue by halting charged particles at the end of their range. Though numerous clinical trials have compared outcomes between the two modalities [3,4], the debate over their superiority persists, as it depends on factors such as disease site, machine parameters, and plan quality [5]. Ultimately, in ideal conditions, the choice between photon and proton therapy is determined by comparing 3D dose distributions from the patient’s planning CT (pCT) [6]. NTCP models are often employed to predict radiation-induced side effects and guide treatment decisions [7].

Clinical decision-making in treatment modality selection is complex [8] due to limited clinical evidence comparing photon and proton radiotherapy [9]. Several studies observed the dosimetric advantages for proton therapy over photon therapy [10,11]. However, a recently randomized clinical trial for non-small-cell lung cancer found no benefit in radiation pneumonitis or local failure with passive scattering proton therapy, despite improved heart dose-volume metrics [12]. An NTCP-based model was proposed to predict cardiac toxicity and facilitate patient stratification [13]. These efforts happen downstream from the initial patient–physician consult, since the dosimetric benefits and toxicity prediction are based on pCT simulation. Another concerning factor is that once the proton modality is selected, insurance approval can take an average of three weeks, worsening if appeals are needed, reported by an institute located in the United States [14]. These prolonged insurance-related delays [15,16] in proton therapy can be associated with increased local recurrence [17] and poor survival [18,19].

Diagnostic CT (dCT) scans, though readily available, may not be directly used for treatment planning due to differences in patient positioning and lack of CT-Hounsfield unit (HU) calibration for dosimetry. One such factor is that the patient anatomy changes from laying on a curved bed in dCT to laying on a flat treatment couch in radiation treatment. The HU histogram can be adjusted and shifted [20,21] to reduce the HU difference in different scanning settings. Significant dosimetric differences have been observed between treatment plans recalculated on pCT and dCT for lung cancer [22]. Despite these obstacles, there have been efforts to develop workflows that utilize dCT to expedite the treatment of critically ill patients and in palliative settings [23]. With the advancements in artificial intelligence (AI), it is now feasible to predict a synthetic CT (sCT) from dCT [22,24,25]. AI-predicted sCT has great potential for improving clinical workflow efficiency, and there are areas yet to be explored.

This study aims to develop a dCT-based workflow for expedited clinical decision-making and modality selection prior to CT simulation using the thoracic site as a proof-of-concept. Three specific aims in this study are: (1) to generate sCT from dCT for thoracic disease site using a DL network; (2) to validate sCT image accuracy in facilitating plan selection between photon and proton treatment modality; and (3) to validate the consistency between two CTs.

## 2. Materials and Methods

### 2.1. Synthetic-CT Workflow for Informed Physician–Patient Consultation

Figure 1 illustrated two workflows, including traditional physician–patient consultation and our proposed modality comparison informed workflow utilizing dCT. This proposed workflow first transforms dCT to sCT using a DL network, followed by manual proton and photon plan generation on sCT for dose and toxicity comparison. In our proposed clinical workflow, physicians determine the optimal treatment modality based on the dose and toxicity comparison generated from the plans created immediately after dCT acquisition.

### 2.2. Network Architecture

A modified 3D U-Net architecture was implemented in MONAI for predicting sCT from dCT [26]. Model parameters were initialized with spatial dimensions set to 3, input channel to 1, extraction level to 3, and 32 initial channels. The network architecture comprises seven encoding and decoding layers along with a bottom block, each with 3D convolutional kernels, batch normalization, and nonlinear activation units [22]. dCT and pCT images were resampled to a uniform resolution of 128 × 128 × 64, serving as the training data and the ground truth, respectively, as shown in Figure 2a. The output block generates the image deformation vector field (DVF), maintaining the same spatial size as the input. The predicted sCT image was then computed using the output DVF and dCT via a warp module [27]. The loss function utilized in finding the optimal DVFs employs the structural similarity index measure loss (SSIMLoss) [26] combined with a regularization term [22]. The generated DVFs are then applied to the input dCT images as part of the training to generate sCT.

#### 2.2.1. Image Pre-Processing and Training

To enhance sCT prediction quality, we developed a couch removal algorithm to eliminate the impact of different CT couches on accuracy using thresholding and morphological operation. Additionally, look-up tables were established for HU value normalization with histogram matching technique [28,29] to minimize variability in dCT scanning energies and protocols. Before the resampling, dCT was rigidly registered to pCT using commercial software, MIM (version 7.2.7, MIM Software Inc., Cleveland, OH, USA), ensuring matching pixel size and slice thickness.

An open-access dataset [30,31] of 46 thoracic cases (dCT and pCT from 4DCT average scans) from two external institutions was used for training (37 cases) and validation (9 cases). An example of scanning parameters can be seen in Table 1 for the public dataset. Both dCT and pCT had arms-up positions for consistency. Using a public dataset enhances model generalizability and facilitates adoption by other hospitals. We trained the model with the Adam optimizer (learning rate of 1 × 10^−5^), with a batch size of 2 and 1500 epochs. After training, 15 cases (input/output size: 512 × 512) with a total of 2648 CT slices were used for treatment planning. The training was performed on a single NVIDIA A10 GPU with 24 GB of memory. The total training time was approximately 30 h. Fifteen patients with arms up and tumors located in the upper lung (with minimal target motion) were selected from our own institution, with detailed scanning parameters shown in Table 1.

#### 2.2.2. Image Evaluation Metrics

MAE and Universal Quality Index (UQI) were used to evaluate the accuracy of the predicted CT images [32]. MAE quantifies the deviation in HU between images, indicating prediction error. UQI models image distortion, ranging from −1 to 1, with 1 representing perfect quality.

The predictive accuracy of our method was benchmarked against both the original pCT images and deformed CT images, named MdCT, generated from a commissioned and validated commercial software, MIM. This comparison provided a first-degree assessment of sCT prediction performance.

### 2.3. Treatment Planning for Dosimetric Assessment

To validate the dosimetric accuracy of the generated sCT, treatment plans were created for both sCT and pCT to compare the dose deviation. For all fifteen thoracic cases selected for testing, the corresponding AI-predicted sCT and ground truth pCT were imported into the Eclipse treatment planning system (Varian Medical Systems, Inc., Palo Alto, CA, USA) for creation and re-optimization of comparison plans on each case. Two-arc volumetric modulated arc therapy (VMAT) technique was used for photon plans, and intensity modulated proton therapy (IMPT) via pencil beam scanning (PBS) was used for proton plans with energies ranging from 10 to 250 MeV. The Non-Linear Universal Proton Optimizer (NUPO) was employed for PBS optimization. The clinical target volumes (CTV) from the pCT images were deformed to dCT with the CT-to-CT adaptive recontour module in a commercial software (MIM). The dCT target volumes were then deformed to sCT with the deformation vector field generated from the network. Clinically acceptable VMAT and PBS plans were created by the same planning expert following target prescription and organ at risk (OAR) constraints, resulting in 60 plans (four per case). Detailed treatment parameters for these original plans are listed in Appendix A.

The dose-volume points obtained from the proton and photon plans were compared to determine the preferable treatment modality. $\Delta {DVH}_{p-ph}^{sCT}$, $\Delta {DVH}_{p-ph}^{pCT}$, and $\Delta {DVH}_{p-ph}^{sCT-pCT}$, shown in Equation (1), represent the DVH deviation between proton and photon plans created using sCT and pCT, respectively. Ideally, $\Delta {DVH}_{p-ph}^{sCT}$ and $\Delta {DVH}_{p-ph}^{pCT}$ will show the same math sign, implying consistent modality preference deduced from sCT and pCT. The investigated DVH constraints included esophagus (mean dose, V35Gy, V60Gy), total lung (mean dose, V10Gy, V20Gy), and heart (mean, maximum dose, V30Gy), all critical OARs in thoracic cancer.(1)$$\left\{\begin{array}{c} \Delta {DVH}_{p-ph}^{sCT}={DVH}_{proton}^{sCT}-{DVH}_{photon}^{sCT}\\ \Delta {DVH}_{p-ph}^{pCT}={DVH}_{proton}^{pCT}-{DVH}_{photon}^{pCT}\\ \Delta {DVH}_{p-ph}^{sCT-pCT}=\Delta {DVH}_{p-ph}^{sCT}-\Delta {DVH}_{p-ph}^{pCT} \end{array}\right.$$

The Concordance Correlation Coefficient (CCC) [33] was utilized to evaluate the agreement between DVH values derived from sCT and pCT, providing an alternative assessment. CCC values above 0.99 indicate near-perfect agreement, 0.95 to 0.99 suggest substantial agreement, 0.90 to 0.95 indicate moderate agreement, and values below 0.90 reflect poor concordance [33].

### 2.4. NTCP Calculation for Treatment Modality Selection

To determine if the advantage treatment modality is consistent between sCT and pCT, NTCP is used for assessing treatment plan quality for critical OAR toxicities, as shown in Figure 2b. For each plan, the NTCP was calculated for three acute endpoints: heart pericarditis, lung pneumonitis, and esophagus perforation. We employed the well-known Lyman Kutcher Burman (LKB) model [34,35] for NTCP calculation. The parameters from the literature [36] for our NTCP calculation are detailed in Appendix A.

The NTCP deviations between the proton and photon plans for each CT image were calculated and are represented by $\Delta {NTCP}_{p-ph}^{sCT}$ and $\Delta {NTCP}_{p-ph}^{pCT}$ in Equation (2). For treatment modality selection, the proton/photon comparison trends (the sign symbol alignment between $\Delta {NTCP}_{p-ph}^{sCT}$ and $\Delta {NTCP}_{p-ph}^{pCT}$) were studied. For example, if the NTCP of photon plan is smaller than that of proton plan for both sCT and pCT, we assume that the trend is same and consistent. CCC and absolute NTCP discrepancy depicted by $\Delta {NTCP}_{p-ph}^{sCT-pCT}$ were analyzed to quantitatively demonstrate the NTCP agreement between the two CT images. We only considered the cases and organs that had NTCP values greater than 0.05%, as the others have a negligible effect on the treatment modality determination.(2)$$\left\{\begin{array}{c} \Delta {NTCP}_{p-ph}^{sCT}={NTCP}_{proton}^{sCT}-{NTCP}_{photon}^{sCT}\\ \Delta {NTCP}_{p-ph}^{pCT}={NTCP}_{proton}^{pCT}-{NTCP}_{photon}^{pCT}\\ \Delta {NTCP}_{p-ph}^{sCT-pCT}=\Delta {NTCP}_{p-ph}^{sCT}-\Delta {NTCP}_{p-ph}^{pCT} \end{array}\right.$$

## 3. Results

### 3.1. Qualitative Comparison of AI-Predicted sCT and pCT

The performance of our AI predicted sCT was compared with the pCT and the MIM deformed MdCT. Figure 3 illustrates two noteworthy examples. The proposed sCT algorithm successfully adapted the body anatomy scanned with a curved couch to one on a flat couch, as observed in Figure 3d. Lung, muscle, and subcutaneous adipose tissue regions predicted by the AI-generated sCT closely matched those observed in the pCT and MdCT. HU profiles in Figure 3e,f demonstrate high HU similarity between the sCT and pCT. Overall, a strong concordance was observed between the AI-generated sCT and pCT. However, red dashed boxes in Figure 3 highlight regions of discrepancies observed in areas involving bony anatomy or contrast-enhanced structures. Since the contrast was only administered during dCT scans and not present in pCT, the AI model performance was limited and could not generate accurate features in sCT. A significant failure where contrast was administered during diagnosis can be also seen in Figure 4. Since there is no contrast in pCT, this contrast will cause dosimetric deviation between sCT and pCT. Additionally, AI failed to predict sCT accurately for dCT with truncation (cut-off in the image), shown in Figure 4 in the red dashed box. These cases with truncation were excluded for dosimetric comparison.

### 3.2. Quantitative Comparison Between AI-Predicted sCT and Commercial Algorithm Deformed MdCT

The accuracy of model performance is validated by comparing the HU deviation between sCT and pCT. The quantitative comparisons between sCT-vs-pCT and MdCT-vs-pCT are presented in Table 2. The MAE is 38.93 ± 14.79 HU between sCT and pCT with an image similarity index of 0.84. The MAE (over all pixels) associated with sCT-vs-pCT was observed to be marginally higher than that for MdCT-vs.-pCT, albeit a smaller standard deviation. In terms of image similarity metrics, the UQI for AI prediction is found to be slightly lower than those for commercial deformation techniques, with a slightly higher standard deviation. The average differences between the two comparisons were 19.38 HU and 0.06 for MAE and UQI, respectively.

MdCT: MIM deformed CT; pCT: planning CT, sCT: AI-predicted synthetic CT.

### 3.3. DVH Metric Comparison Between sCT and pCT for Selected Critical OARs

An illustrative DVH example of optimized plans derived from both pCT and sCT is presented in Figure 5a,b. The esophagus V35Gy of the proton plan is lower than that of the photon plan for both pCT and sCT, indicating a consistent trend between the two types of CT images. The magnitude of this V35Gy improvement, highlighted by the red arrows, between the proton and photon plans on each image was further examined to assess the accuracy of the AI-predicted sCT in comparison to the pCT. A detailed comparison (between photon and proton plans) of DVH metrics derived from both sCT and pCT is shown in Figure 5c. Based on these results, the proton plans were preferred for 39 DVH values and the photon plan was preferred for 4 DVH values of both sCT and pCT. Disagreement between plans on sCT and pCT was revealed in only 4 DVH values, as pointed out by red circles in Figure 5c. This resulted in a high total agreement rate of 91.5% in modality selection between sCT- and pCT-based treatment plans. All points are distributed around the blue line, and the CCC for these DVH values was 0.90. This shows a moderate dosimetric agreement between sCT and pCT. Figure 5d presents a quantitative comparison of DVH values between pCT and sCT, revealing a mean absolute deviation of 3.52%, 3.30%, and 3.60% for the esophagus, total lung, and heart, respectively. Notably, the total lung V10Gy and V20Gy exhibited the largest dispersion in DVH value deviation between sCT and pCT.

### 3.4. NTCP Alignment and Quantitative Deviation Between sCT and pCT

Out of 45 NTCP values considered in our study, 19 were selected for the final comparison based on organ sensitivity (NTCP > 0.05%). The overall trend alignment for these selected OAR NTCP values is depicted in Figure 5e. Our analysis on pCT revealed that 12 NTCP values decreased for proton plans in comparison to photon plans, while 7 NTCP values increased. Remarkably, our expedited workflow using sCT predicted these same trends with 100% accuracy. The CCC for the NTCP comparison was 0.97, which shows substantial agreement between sCT and pCT in terms of OAR toxicity. The absolute NTCP deviation between sCT and pCT, $\Delta {NTCP}_{p-ph}^{sCT-pCT}$, was 1.54%, 0.21%, and 2.36% for esophagus perforation, lung pneumonitis, and heart pericarditis, respectively, as shown in Figure 5f. Furthermore, the median NTCP discrepancies were minimal, calculated to be −0.02%, −0.01%, and 0.03% for these three organs, respectively.

## 4. Discussion

We propose an AI-powered workflow to provide physicians with dose and toxicity comparisons between photon and proton treatments for informed decision-making during initial consultations. This proof-of-concept study using sCT enables faster access to dose distribution and NTCP comparisons without waiting for pCT acquisition. For critically-ill patients in need of a fast treatment turnaround, the treatment plan generated from sCT can be delivered with cone beam CT (CBCT) image-guided verification or CBCT-enabled plan adaptation [37]. While promising, the current workflow has constraints that may limit immediate clinical implications. Our model was trained on dCT scans with arms-up position and no truncation. Consequently, cases with discrepancies in arm positioning, such as arms-down in pCT, were excluded. Additionally, to ensure tumor consistency between dCT and pCT, we manually excluded cases with substantial tumor shape change. These aspects may present challenges in clinical deployment. Currently, our study serves as a proof-of-concept under research conditions.

Results from the study showed that AI-generated sCT had an MAE of 38.93 HU and a UQI of 0.84 as compared to pCT. Studies have demonstrated that such AI-predicted sCTs can be effectively utilized for treatment planning to facilitate expeditious treatment [38]. In comparison, previous studies reported an MAE of 54.9 HU for sCT generated from MR images for lung cancer, achieving good dose accuracy with mean gamma rates of 98.2% (3%/3 mm) [38]. Thus, our MAE results indicate high validity of the AI-generated dCT-to-pCT synthesis. With both dCT and pCT as the input, the MdCT generated from the commercial approach outperformed our AI-prediction by 19.38 HU in MAE. This is because the commercial software has knowledge of both pCT and dCT, whereas the sCT was directly derived from dCT through a DL approach with no input from pCT. A similar trend was observed that the Elastix image registration algorithm surpassed a 3D convolutional neural network in body Dice similarity coefficient (DSC) and root averaged squared sum of differences (RASSD) [24]. Furthermore, the performance of AI synthesis of sCT depends on the size of the training dataset. The performance of U-Net was reported to be suboptimal in a dCT-to-pCT synthesis study using 10 cases for training [23]. Alternatively, 40 cases [38] and 64 cases [39] were utilized for MR-to-pCT synthesis in previous studies, which presented an acceptable image quality and dose accuracy. To our best knowledge, our study is the first to use an external dataset of 46 cases for DL network training, achieving a reasonable image similarity for testing cases from our institution.

DVH metrics between photon and proton plans showed 91.49% agreement between sCT and pCT, with four misaligned metrics likely due to minor HU variations and manual optimization. Treatment modality comparison was also evaluated based on toxicity modeling. NTCP results based on sCT fully aligned with those based on pCT. NTCP differences for heart pericarditis were consistent, with $\Delta {NTCP}_{p-ph}^{sCT}$ and $\Delta {NTCP}_{p-ph}^{pCT}$ at −1.78% and −6.16%, respectively, favoring the proton plan. The maximum NTCP discrepancy was 2.36%, likely due to HU errors and optimization. While treatment selection based on NTCP was accurate, small changes in DVH near tolerance doses can lead to significant NTCP deviations, as seen in Figure 5e, where minor DVH differences caused notable NTCP variations due to proximity of the OAR to the target.

U-Net has been widely adopted in medical image segmentation and image generation tasks due to its strong performance with limited training data [40]. In our study, we selected U-Net for the development of the synthetic CT generation model because of its relatively simple structure and potential generalization with limited training data. We acknowledge the rapid progress in deep learning architectures, such as Transformer-based models, which demonstrate promising results in image synthesis. These state-of-the-art models will be explored in future studies to compare their synthetic CT generation performance with that of U-Net, especially as more training data are made available.

There are a few notable limitations in our study. First, variability in manual plan creation on pCT and sCT may have affected the target coverage vs. OAR sparing trade-offs, despite using the same plan settings and DVH constraint weights. To mitigate this variability, we focused on the qualitative comparison trends between proton and photon plans derived from the two CTs. Future study can adopt automatic treatment planning tools for less variability in manual planning. Additionally, while the training data size was adequate for this preliminary study, it may not suffice for clinical use. We utilized data augmentation techniques such as rotating images clockwise by 45 degrees and flipping images horizontally [41] to increase the training data size. This workflow has the potential to allow attending physicians to determine the most ideal treatment modality during the initial patient–physician consultation and submit insurance claims immediately after dCT acquisition. However, model performance should be evaluated with larger datasets for future clinical applications. CT-HU calibration is not acquired in this study, but we employed the HU histogram shifting [20] to mitigate the effect of different scanning settings. Finally, this proof-of-concept study explores the feasibility of employing sCT for treatment modality selection. Due to the absence of 4D-CT during the diagnostic stage, robust optimization for PBS plans is considered outside the scope of this study.

## 5. Conclusions

We developed an expedited AI-powered workflow converting dCT to sCT for photon and proton plan comparison and treatment modality selection. To our knowledge, this is the first study testing the feasibility of using DL-based sCT for shortening the overall patient wait time.

## Figures and Tables

**Figure 1 cancers-17-01553-f001:**
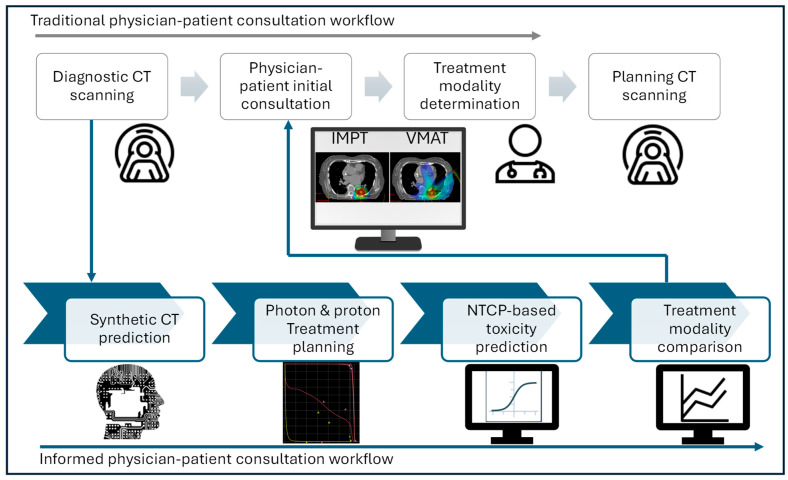
Informed physician–patient consultation workflow with synthetic CT.

**Figure 2 cancers-17-01553-f002:**
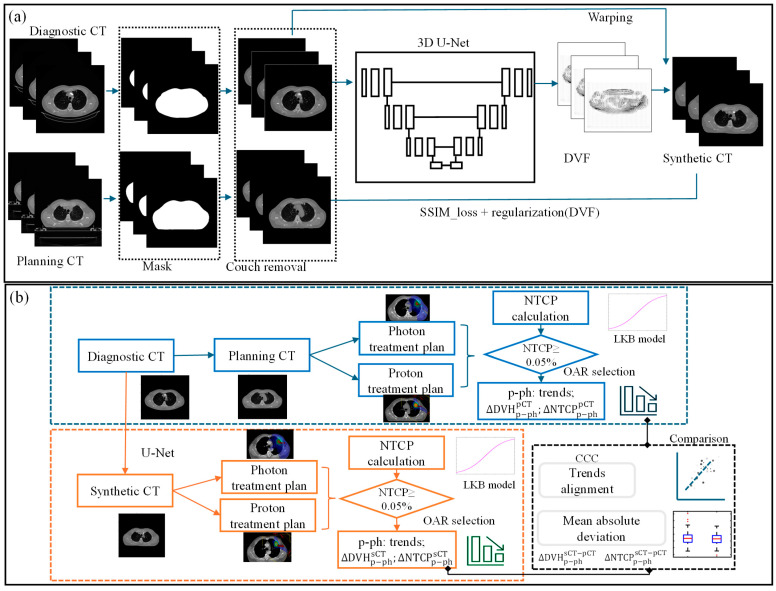
(**a**) Workflow of U-Net-based synthetic CT prediction from diagnostic CT and (**b**) the validation process for the NTCP-based treatment modality selection with sCT ($\Delta {DVH}_{p-ph}^{sCT}$, $\Delta {DVH}_{p-ph}^{pCT}$ and $\Delta {DVH}_{p-ph}^{sCT-pCT}$ represent the DVH deviation between proton and photon plans created using sCT and pCT, and the DVH value agreement between two CTs, respectively. The NTCP deviations between the proton and photon plans for each CT image were calculated and are represented by $\Delta {NTCP}_{p-ph}^{sCT}$ and $\Delta {NTCP}_{p-ph}^{pCT}$. Their agreement is denoted by $\Delta {NTCP}_{p-ph}^{sCT-pCT}$).

**Figure 3 cancers-17-01553-f003:**
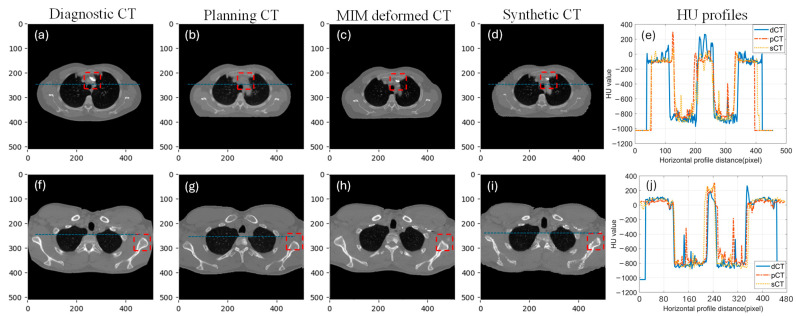
Two example cases with each column showing dCT, pCT, MdCT, sCT, and HU line profile taken from the blue line on dCT, pCT, and sCT. ((**a**–**e**): case 1, (**f**–**j**): case 2, red dash box highlights the difference between diagnostic CT, planning CT, MIM deformed CT and synthetic CT).

**Figure 4 cancers-17-01553-f004:**
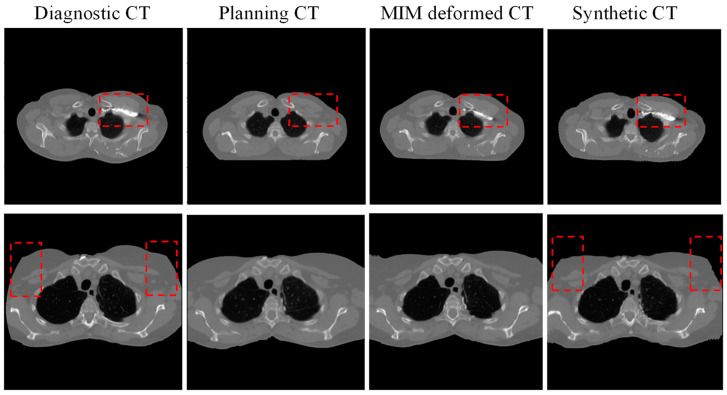
Two examples of failed prediction of synthetic CT (from left to right: diagnostic CT, planning CT, deformed CT with commercial software MIM, and synthetic CT. red dash box highlights the significant difference between diagnostic CT, planning CT, MIM deformed CT and synthetic CT).

**Figure 5 cancers-17-01553-f005:**
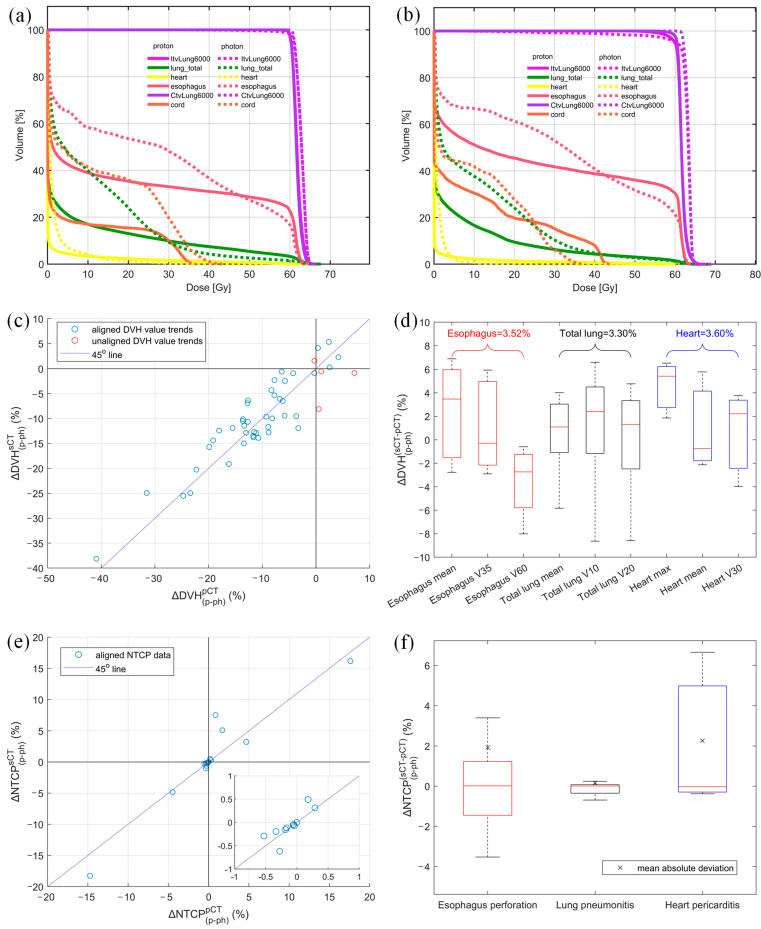
DVH comparison of proton and photon plan derived from (**a**) planning CT and (**b**) AI-predicted synthetic CT, (**c**) DVH value alignment, (**d**) absolute DVH value discrepancy between sCT and pCT for each structure with the mean absolute deviation shown above each structure, (**e**) NTCP alignment, and (**f**) absolute NTCP discrepancy between proton and photon plans derived from sCT and pCT (mean absolute NTCP variation between two CTs shown in × sign. Different colors represent different endpoints).

**Table 1 cancers-17-01553-t001:** Scanning parameters of dCT and pCT.

**Scanning Parameters**	**Public Data**		**Institutional Data**		
	**dCT**	**pCT**	**dCT**		**pCT**
Manufacturer	GE	Philips	Philips	GE	Siemens
Convolution kernel	SOFT	B	B	-	Br38s
kVp (kV)	120	120	120	120	120
Data collection diameter (mm)	500	500	500	500	500
X ray tube current (mA)	296	297	333	232	177
Exposure (mAs)	20	250	77	15	221
Filter type	Body filter	B	B	Standard	FLAT
Pixel spacing (mm)	0.9766	0.9766	0.9766	0.9766	0.9766

**Table 2 cancers-17-01553-t002:** Quantitative comparison of MIM deformed MdCT and AI-predicted sCT with pCT.

**Metric**	**MdCT vs. pCT**	**sCT vs. pCT**
MAE	19.55 ± 3.78 HU	38.93 ± 14.79 HU
UQI	0.90 ± 0.02	0.84 ± 0.05

## Data Availability

Research data will be shared upon request to the corresponding author.

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
