# Peer review of "Deep Learning-Based Synthetic CT for Personalized Treatment Modality Selection Between Proton and Photon Therapy in Thoracic Cancer"

_cancers, 2025, doi:10.3390/cancers17091553_

Round 1

Reviewer 1 Report

Comments and Suggestions for Authors

<Major Comments>

  1. Novelty and Significance
  • The idea of using DL-predicted sCT for treatment modality comparison at the consultation stage is innovative and clinically meaningful.
  • The study convincingly shows high agreement between sCT-based and pCT-based planning in terms of DVH metrics and NTCP analysis.
  • Nevertheless, the authors should better emphasize the potential clinical implications and limitations for real-world application in the Discussion, especially regarding regulatory or safety concerns.
  •  
  1. Validation Cohort and Generalizability
  • The model was trained on 46 external cases but validated on only 15 institutional cases.
  •  Please clarify if the external dataset characteristics (e.g., scanner types, acquisition protocols) differ significantly from the institutional validation dataset, which could impact generalizability

  1. Validation Cohort and Generalizability

-The model was trained on 46 external cases but validated on only 15 institutional cases.

- Please clarify if the external dataset characteristics (e.g., scanner types, acquisition protocols) differ significantly from the institutional validation dataset, which could impact generalizability

<Minor point>

  1. Ethics Statement
  • The IRB protocol code is listed as "XXXX." → Please update with the actual approved protocol number before final publication.
  1. Citations and References
  • Some references are repeated (e.g., ref 40 and 41 seem to refer to the same study). → Ensure references are unique and correctly formatted.

Reviewer 2 Report

Comments and Suggestions for Authors
  1. The Simple Summary aims for accessibility but introduces technical terms like VMAT, PBS, and NTCP early on, which may be unfamiliar to readers outside the immediate radiation oncology/medical physics community. Consider beginning with a higher-level statement of the clinical problem (e.g., difficulty in selecting the best radiation therapy type before dedicated planning scans) and the core idea of the proposed solution (using AI to create a usable 'synthetic' CT from an earlier diagnostic CT) before diving into specific acronyms or techniques.
  2. While the abstract and introduction lay out the study, ensuring the primary objective and the specific contribution are exceptionally clear throughout the manuscript would strengthen its impact.
  3. Missing space: "pointed outby" (Line 220) should be "pointed out by".
  4. Spelling error: "distrubuted" (Line 222) should be "distributed".
  5. Several relevant acronyms and units appear without being defined upon first use (e.g. OAR, HU, CBCT,...). While many core acronyms are defined correctly, this inconsistency needs addressing.
  6. The manuscript employs a significant number of acronyms. While these are generally defined, their high density throughout the text can sometimes interrupt reading flow and increase cognitive load for the reader. The authors are encouraged to review the necessity of each acronym, particularly those used less frequently, and consider reducing the overall count where possible to enhance readability.
  7. The method for transferring target volumes (ITV/CTV) from pCT/dCT to the sCT needs more detail. It states they were deformed "with our trained U-Net network", but U-Net typically generates images, not contour deformations directly.
  8. The results section should be extended to support multiple scenarios. The authors should better explain the performance (e.g. the results of Fig. 3) and also provide examples where their model failed. Specific recommendations:
    1. Providing a more detailed interpretation of the visual results in Figure 3, discussing both the successes and the highlighted discrepancies in bony or contrasted anatomy.
    2. Adding examples illustrating limitations or potential failure modes of the sCT generation or the downstream comparison process. This offers a more comprehensive performance evaluation.
  9. The Conclusion states the workflow "allows attending physicians to determine the most ideal treatment modality... and submit insurance claims immediately after dCT acquisition." While this is the goal, the study only demonstrates feasibility of comparison on sCT. It doesn't provide evidence that this workflow directly enables immediate, confident clinical decisions or impacts insurance submission times in practice.
  10. HW: Specify the hardware configuration (e.g., GPU model(s), number of GPUs, RAM) employed for both the training process and for running inference. Also include information about the computational time required for training, such as the approximate time per epoch or the total training duration.
  11. Train/val/test: Clarify precisely how the datasets were split. The text mentions training on a public dataset (46 cases) and testing on an institutional one (15 cases). Was the public dataset further split into training and validation sets? Please specify the number of cases used for training vs. validation before final testing.
  12. Line 133: (xxxx) seems to be a placeholder.
  13. The study utilizes the U-Net architecture. However, given the rapid advancements in deep learning, particularly the rise of Transformer-based architectures (e.g., Vision Transformers, Swin Transformers) for vision tasks since its publication, it would strengthen the paper to briefly address this. Could the authors comment on the rationale for choosing U-Net for this specific dCT-to-sCT application? A short discussion point on why U-Net was considered appropriate, or acknowledging the potential of exploring newer architectures like Transformers as a promising direction for future work, would provide valuable context regarding the current state-of-the-art.

Round 2

Reviewer 2 Report

Comments and Suggestions for Authors

The conclusion part can be expanded.